# Characterizations of the viability and gene expression of dispersal cells from *Pseudomonas aeruginosa* biofilms released by alginate lyase and tobramycin

Said M. Daboor[1,2], Renee Raudonis[1], Zhenyu Cheng[1]*

**1** Department of Microbiology and Immunology, Dalhousie University, Halifax, Nova Scotia, Canada,
**2** National Institute of Oceanography and Fisheries, Cairo, Egypt

* zhenyu.cheng@dal.ca

**Data Availability Statement:** All relevant data are within the manuscript and its Supporting Information files.

## Abstract

Biofilm infections are hard to manage using conventional antibiotic treatment regimens because biofilm structures discourage antibiotics from reaching the entire bacterial community and allow pathogen cells to persistently colonize and develop a plethora of tolerance mechanisms towards antibiotics. Moreover, the dispersed cells from biofilms can cause further complications by colonizing different sites and establishing new cycles of biofilms. Previously, we showed that alginate lyase enzyme (AlyP1400), purified from a marine *Pseudoalteromonas* bacterium, reduced *Pseudomonas aeruginosa* biofilm biomass and boosted bactericidal activity of tobramycin by degrading alginate within the biofilm extracellular polymeric substances matrix. In this work, we used a flow cytometry-based assay to analyze collected dispersal cells and demonstrated the synergy between tobramycin with AlyP1400 in enhancing the release of both live and dead biofilm cells from a mucoid *P. aeruginosa* strain CF27, which is a clinical isolate from cystic fibrosis (CF) patients. Interestingly, this enhanced dispersal was only observed when AlyP1400 was combined with tobramycin and administered simultaneously but not when AlyP1400 was added in advance of tobramycin in a sequential manner. Moreover, neither the combined nor sequential treatment altered the dispersal of the biofilms from a non-mucoid *P. aeruginosa* laboratory strain PAK. We then carried out the gene expression and tobramycin survival analyses to further characterize the impacts of the combined treatment on the CF27 dispersal cells. Gene expression analysis indicated that CF27 dispersal cells had increased expression in virulence- and antibiotic resistance-related genes, including *algR*, *bdlA*, *lasB*, *mexF*, *mexY*, and *ndvB*. In the CF27 dispersal cell population, the combinational treatment of AlyP1400 with tobramycin further induced *bdlA*, *mexF*, *mexY*, and *ndvB* genes more than non-treated and tobramycin-treated dispersal cells, suggesting an exacerbated bacterial stress response to the combinational treatment. Simultaneous to the gene expression analysis, the survival ability of the same batch of biofilm dispersal cells to a subsequent tobramycin challenge displayed a significantly higher tobramycin tolerant fraction of cells (~60%) upon the combinational treatment of AlyP1400 and tobramycin than non-treated and tobramycin-treated dispersal cells, as well as the planktonic cells (all below 10%). These results generate new knowledge

**Funding:** The study was supported by the Cystic Fibrosis Canada, Institute of Infection and Immunity (PJT165970) and Nova Scotia Health Research Foundation. The funders had no role in study design, data collection and analysis, decision to publish, or preparation of the manuscript.

**Competing interests:** The authors have declared that no competing interests exist.

about the gene expression and antibiotic resistance profiles of dispersed cells from biofilm. This information can guide the design of safer and more efficient therapeutic strategies for the combinational use of alginate lyase and tobramycin to treat *P. aeruginosa* biofilm-related infections in CF lungs.

## Introduction

*Pseudomonas aeruginosa* is a ubiquitous bacterial pathogen that is responsible for the high morbidity and mortality rates of CF patients [1]. Biofilm formation is a common strategy used by successful pathogens to combat antimicrobial and immune responses [2,3]. For biofilm-related *P. aeruginosa* infections, bacterial cells are enclosed within an extracellular polymeric substances matrix, consisting mainly of polysaccharides, proteins, extracellular DNA and lipids, which can function as an antibiotic barrier [4–6]. Biofilm formation is a dynamic multi-step process, cycling between initial attachment, to expansion, then to final dispersal of the biofilm [7,8]. Each of these steps contains diverse bacterial cell populations that are characterized by a unique physiological status that differs from the planktonic free-living cells [9,10]. Dispersal cells represent a distinct population [11]. These cells are important for the dynamic cycling of biofilm formation. It is thought that the biofilm dispersed cells immediately undergo physiological transitions and enter the planktonic phase. However, Chua et al. showed that the dispersal of *P. aeruginosa* correlated with a specific dispersal phenotype that was largely different from those of planktonic or biofilm cells [11]. For example, the type 2 secretion system (T2SS) genes were at least fivefold upregulated in dispersed cells compared with planktonic cells.

Many anti-biofilm strategies have been developed in order to tackle the biofilm-related infection problems. One promising approach is the use of enzyme to target and degrade alginate, the major polysaccharide component within *P. aeruginosa* biofilm [12–16]. Alginate lyases manifest biofilm-dispersive properties and display synergy with clinically relevant antibiotics to disrupt *P. aeruginosa* biofilms and improve the anti-pseudomonal antibiotic efficacy [12–16]. Previous anti-biofilm studies primarily focussed on addressing the question of how various biofilm disrupting agents affected the final outcome of remaining biofilms. Thus, there is a lack of characterization of the effects on dispersing biofilm cells and little is known about the viability and physiological changes of these dispersal cells. In this study, we directly captured the dispersed cells under various treatments to quantify both viable and non-viable bacterial cell events using a flow cytometric analysis and to characterize the expression profiles of genes involved in virulence and drug-resistance in the dispersal cells. In summary, our data filled the knowledge gap in the characterizations of viability and gene expression patterns of dispersal cells from biofilms. Surprisingly, large quantities and percentages of the dispersal population were viable and demonstrated a gene expression pattern that is associated with higher virulence compared to planktonic cells. More importantly, a higher percentage of the dispersal cells from the combinational treatment survived sub-lethal doses of tobramycin. This knowledge can better guide the future design of therapeutic strategies utilizing the combination of biofilm disruptive agents with antibiotics to treat *P. aeruginosa* lung infection in CF patients.

## Materials and methods

### *P. aeruginosa* biofilm cultivation and treatments

The cultivation of *P. aeruginosa* mucoid CF isolate CF27 [17] biofilm in a flow cell continuous system ($1 \times 40 \times 44$ mm; Biocentrum, DTU, Denmark), including the cultivation and

treatment conditions, was optimized and described in our previous studies [15,16]. The flow cell biofilm system is a robust method for mimicking the respiratory tract of CF patients. Briefly, an overnight CF27 culture was centrifuged and washed three times in PBS. Cells were resuspended in M63 medium (for 1L, 13.6 g $KH_2PO_4$, 2 g $(NH_4)_2SO_4$, 0.2 g $MgSO_4 \cdot 7H_2O$, 2 g glycerol, 0.5 mg $FeSO_4 \cdot 7H_2O$, 0.5 mg vitamin B1, and 1.0 mg L-arginine, pH 7.0) and adjusted to $OD_{600}$ of 0.5. The three-channel flow cell was assembled and sterilized following the manufacturer's instructions. Each of the three channels of the flow cell was injected with 250 μL of CF27 cell suspension, as prepared above. The flow cell was incubated statically for two hours at 37˚C. The biofilms were developed on the glass substratum on top of the flow cell with 96 h of incubation at 37˚C with the continuous flow of M63 medium (0.2 mm/s linear flow rate). Based on our previous optimization [15], the 96-hour old biofilm showed high stability and consistency between different batches, enabling a reliable comparison of the effects of various treatments on dispersals from uniformly grown initial biofilms. After the establishment of 96-hour old mature biofilm, the flow cell was incubated at a static condition for 30 minutes before each channel was injected with 250 μL of one the following: M63 medium (non-treated control), 2× minimum inhibitory concentration (MIC) tobramycin (16 μg/mL), AlyP1400 (250 U/mL) simultaneously with tobramycin (16 μg/mL) (referred to as the combinational treatment, shown as AlyP1400+TOB)), or AlyP1400 (250 U/mL) followed by tobramycin (16 μg/mL) (referred to as the sequential treatment, shown as AlyP1400→TOB). Each flow cell was kept under static conditions for another 2 hours before the flow of M63 medium was resumed for the collection of dispersed biofilm cells. For the sequential treatment, AlyP1400 was injected into the biofilms and incubated for 2 hours followed by tobramycin injection and incubation for another 2 hours. One mL aliquots of biofilm effluent runoff were collected at 0, 3, 6, 12 and 24 hours after the resumption of the flow following the above treatments to analyze the dispersed cells. Samples were centrifuged at 12,096 × *g* for 10 minutes at 4˚C. For enumeration of the colony forming units (CFU), the pellets were resuspended in one mL of M63 and serially diluted 10-fold before plating on LB agar plates.

In parallel, separate flow cells were used in identical set up as above for the confocal microscopic analysis of the remaining biofilms. After the 2-hour treatments, each channel of the flow cell was then stained for 30 min with 250 μL, 10 μM Syto 61 Red (Thermo Fisher Scientific, Invitrogen), followed by 250 μL, 10 μM Sytox Green (Thermo Fisher Scientific, Invitrogen) for another 30 min. The confocal microscopic analysis of the remaining biofilms post-treatments was carried out as previously described [15].

To investigate the effects of AlyP1400 on a non-mucoid strain of *P. aeruginosa*, the flow cell biofilm of lab strain PAK [18] was cultivated and treated as described above for CF27. CFU counting on LB agar plates of the dispersal cells from various treatments were carried out using serially diluted PAK dispersal cells.

## Flow cytometric analysis of dispersed biofilm cells' viability

Dispersal biofilm cells were measured in the collected samples using a CytoFlex flow cytometer (Beckman Coulter, USA), equipped with a 50 mW 488 nm blue laser, a 50 mW 638 nm red laser and an 80 mW 405 nm violet laser. Two fluorescence dyes were used to stain and count the dispersal cells from biofilms. Following the manufacturer's instruction (BD Bioscience, USA), Fixable Viability Stain 520 (FVS520) and FVS700 were used to distinguish the live cells from the membrane-compromised cells. One mL aliquot of biofilm effluent runoff was stained with 1 μL FVS520 to label dead or membrane-compromised cells (hereafter referred to as dead cells), followed by fixation with 90% methanol and then stained with 1 μL FVS700 to label all the cells. Cells were washed with 1% BSA in PBS after each staining and fixation step. Fixed

and stained bacteria were kept at 4˚C in the dark until the samples could be recorded; samples were recorded within 24 hours of staining. The CytoFlex was configured to record violet side scatter (VSSC; according to manufacturer instructions) to better resolve the bacteria from background signals (excitation with 405 nm laser and emission detected using a 405/10 band-pass filter). FVS520 was excited using the 488 nm laser and was detected using an 525/40 emission bandpass filter, while FVS700 was excited using the 638 nm laser and was detected using a 712/25 bandpass filter. Data were collected in logarithmic mode and analysed with FCS Express (De Novo Software, USA). Bacterial cells were identified by gating on an initial plot of VSSC versus forward scatter (FSC), and then gating on all events that were FVS700-positive, as only the bacterial cells should have the staining, thus separating bacterial cell signals from the background (inorganic and organic particles) [19,20]. The number of live and dead cell events were then determined by gating on the FVS520, where cells positive for FVS520 were considered dead or membrane-compromised and FVS520 negative cells were considered live. The CytoFlex was volumetrically calibrated according to manufacturer's instructions prior to acquiring data and the volume measured and recorded for each sample was used to calculate the live and dead cell number per mL.

### Gene expression analysis of dispersed biofilm cells

The dispersed CF27 cells were collected from 96-hour old biofilms cultivated in the flow cell after treatments with medium control, tobramycin alone, or tobramycin combined with the AlyP1400. One mL of dispersed cells was collected directly into 2 mL of RNA Protect Bacteria Reagent (Qiagen, Germany). One mL of CF27 inoculum for the biofilm cultivation was added to 2 mL of RNA Protect Bacteria Reagent as a planktonic cell control for comparison. The bacterial cells were pelleted by centrifugation at 12,096 × g at 4˚C for 10 minutes. RNA was isolated from planktonic and dispersal cells using Trizol reagent [21,22] (Invitrogen, Thermo Fisher, USA) according to the manufacturer's protocol. The residual genomic DNA was removed from purified RNA by DNase treatment using the DNA free kit (Ambion, Thermo Fisher, USA). The concentration and purity of RNA were checked spectrophotometrically with a NanoDrop (Thermo Scientific, Whaltam, MA). All the samples showed an absorbance 260/280 ratio of ≈ 2, their RNA integrity was verified using a 1% bleach agarose gel [23]. cDNA was synthesized from 1 μg RNA using iScript cDNA Synthesis Kit (Bio-Rad, USA), as recommended by the manufacturer's instructions. Samples without reverse transcriptase enzyme were used as negative control for monitoring any genomic DNA contamination. The RT-qPCR primers used for the reference gene (*rpsL*) and the ones for the target genes in this work (Table 1) were either previously published [24,25] or designed using NCBI primer-Blast online tool using *P. aeruginosa* strains PAO1 and PA14 as the reference genomes. The amplification product specificity for each primer was verified with gDNA from CF27 as a template using a standard PCR method [26]. GoTaq® Colourless Master Mix (Promega, Madison, WI, USA) was used for the PCRs following the manufacturer's instructions and contained 0.2 μM of each primer. PCR products with the expected size band were visualized on a 1% agarose gel stained with SYBR® Safe DNA Stain (Invitrogen, Carlsbad, CA, USA) under UV light. The amplification efficiency for each primer pair was calculated as E = $[10^{(-1/slope)}] \times 100\%$, in which the slopes were obtained from the standard curve generated from serial dilutions that have a correlation coefficient of at least 0.98 ($R^2 > 0.98$) of pooled cDNAs [27,28]. The stability of the housekeeping gene was verified by generating Cycle-threshold (Ct) values by the $2^{-\Delta Ct}$ method [28] using the CFX manager™ software (Biorad CFX, USA). The 20 μL RT-qPCR reactions contained 8 μL cDNA (80 ng), or nuclease free water (no-template control), 10 μL Advanced qPCR Master Mix with SUPERGREEN (Wisent, Inc, Canada), and 1 μL of each

**Table 1. Primers for genes to characterize the expression profiles of dispersal cells and the reference gene that were used in this study.**

| Gene | Gene description | Primer sequences (5'→3') | Amplicon length bp | Amplification efficiency % | Regression coefficient | Reference |
|---|---|---|---|---|---|---|
| *algD* | GDP-mannose 6-dehydrogenase for alginate synthesis | CGGTCATGAAGTCATTGGTG AACGATACGTCGGAGTCCAG | 177 | 101.3 | 0.999 | This work |
| *algR* | A transcriptional factor that regulates alginate synthesis | TTCATTGCCGACCACAAGTA TCGAGGCCTTTCAGGTAGAG | 200 | 102.0 | 0.987 | This work |
| *algU* | Extracytoplasmic function sigma factor responsible for alginate overproduction in *P. aeruginosa* | TTTGTCGATTGCTTCACGAG GCGAGTTCGAAGGTTTGAGT | 104 | 108.7 | 0.999 | This work |
| *bdlA* | Chemotaxis transducer protein BdlA (Biofilm dispersion locus A) that controls biofilm dispersion | CTACGCGCAATCGGAAGAC GGACATTGCCGTCGAGGTC | 213 | 103 | 0.97 | 24 |
| *clpX* | ATP-dependent Clp protease ATP-binding subunit | GTGGGCGAGGATGTCGAGAAC CGGTACCCTCGATGAGCTTCAG | 190 | 99.2 | 0.996 | This work |
| *lasB* | Encodes elastase B (LasB), an extracellular protease thermolysin metallopeptidase | TGATCGGCTACGACATCAAG ATTGGCCAACAGGTAGAACG | 161 | 108.5 | 0.98 | This work |
| *lasR* | LuxR family transcriptional regulator. LasR activates transcription of some genes for QS regulation | AAGGACAGCCAGGACTACGA GTAGATGGACGGTTCCCAGA | 156 | 103.6 | 0.988 | This work |
| *mexB* | Resistance-Nodulation-Division (RND) efflux family. Contributes to the intrinsic resistance to aminoglycoside. | CCTGCTGATCTACGTGGTGA CCTTCTCCAGCAGGTATTCG | 182 | 109 | 0.94 | This work |
| *mexF* | Membrane fusion protein, multidrug efflux RND (transporter permease subunit) | TCTACGACCCGACCATCTTC AGGAACAGGATCACCACCAG | 100 | 107 | 0.996 | This work |
| *mexY* | RND efflux family. Contributes to aminoglycoside resistance. | CAACGGCTATCCCTCGTTCA AACACGATCAGCACCGAGAG | 198 | 112 | 0.96 | This work |
| *mexZ* | RND efflux family. Contributes to the intrinsic resistance to aminoglycoside. | TGGCCAGAAAAACCAAAGAG CAGGCAGACCTCGATCTTGT | 179 | 105.7 | 0.999 | This work |
| *ndvB* | Encodes glucosyltransferase enzyme to promote periplasmic β-(1–3) cyclic glucans. Enhances aminoglycoside resistance. | CTGCTGCTGATCGACAGTTC GCTGTAGTCGTAGGCGATCC | 108 | 90.8 | 0.999 | This work |
| *rpsL* | Cell division protein 30S ribosomal subunit protein S12, used here as the reference gene. | GCAAGCGCATGGTCGACAAGA CGCTGTGCTCTTGCAGGTTGTGA | 80 | 96.9 | 0.999 | 25 |

primer (2 μM). The RT-qPCR reactions were performed using the CFX Connect Real-Time thermal cycler (BioRad, California, USA) and completed with a heat activation cycle at 95°C for 3 minutes, followed by 44 cycles of denaturation at 95°C for 10 s, and annealing/extension at 60°C for 25 s with data acquisition. To confirm the specific amplification of a single PCR product, a melting curve was determined over the temperature range 60–95°C at 0.5°C increments [29,30]. The relative levels of gene expression were quantified using the $2^{-\Delta\Delta Ct}$ method, and the fold differences in the expression of target genes between the biofilm dispersed cells and planktonic cells were normalized using *rpsL* as the reference gene. In this study, the planktonic cells are the CF27 inoculum (washed and diluted to $OD_{600} = 0.5$ in M63 medium) used for the biofilm cultivation in the flow cell chambers.

## Tobramycin survival analysis

To investigate whether biofilm dispersal cells released under various conditions had an advantage when grown in the presence of tobramycin, serial dilutions of planktonic inoculum and dispersal cells from biofilms for both CF27 and PAK under different treatments and the non-treated control were plated on LB agar in the absence or presence of four sub-MICs (0.25 to 8 μg/mL, MIC = 16 and 8 μg/mL for CF27 and PAK, respectively) of tobramycin as described previously [31]. For each treatment condition, survival advantage in the presence of sub-MIC

of tobramycin was expressed as the tobramycin tolerant fraction, as determined by the CFU counts from LB plates with tobramycin divided by the CFU counts from LB plates without tobramycin, multiplied by 100.

## Statistical analyses

Analysis was performed using GraphPad Prism software (version 7.0; GraphPad Software, Inc, La Jolla, CA). ANOVA one-way and two-way followed by Tukey's multiple comparison tests were used to determine any statistical significance difference between separate experimental conditions (P values < 0.05 were considered significant).

## Results

### Combined use of tobramycin with alginate lyase increases dispersal of *P. aeruginosa* CF27 biofilms

Mature biofilms generate dispersal cells that can re-attach to a new surface, initiating a subsequent cycle of biofilm formation at distant locations. Previous studies exploring various biofilm disruption strategies had almost exclusively focused on the direct effects on biofilm reduction. Thus, little is known about the downstream events for dispersal cells and the subsequent consequence on biofilm re-initiation. The biofilm dispersal cells represent a unique intermediate step between the planktonic and biofilm life style, and are highly virulent to the host cells [11]. Our previous study demonstrated that the combined use of tobramycin with alginate lyase reduced *P. aeruginosa* CF27 biofilm and enhanced bactericidal activity within biofilms [16]. We repeated the confocal microscopic experiments on the control and treated biofilms that were carried out in our previous work focusing on the effects of treatments on the remaining biofilms [15,16]. The data in this work confirmed our previous finding that the AlyP1400 combined with tobramycin led to a dramatically reduced biomass (as reflected by the decrease in fluorescence intensity) compared to the biofilms that were treated with tobramycin alone, and the confocal images are shown in S1 Fig as a visual confirmation of the biofilm reduction. This substantial difference suggested that the lyase activity of AlyP1400-trigged biofilm disruption is essential for enhancing the bactericidal properties of tobramycin in biofilm eradication.

After verifying the treatment results on the CF27 biofilm matched our previous findings, the focus of this work is to directly quantify the number of dispersal cells and examine their responsiveness to treatments. First, we conducted a flow cytometric bacterial cell viability analysis of dispersed biofilm populations. We evaluated the dispersal cells after treatment with 16 μg/mL of tobramycin individually or tobramycin in combination with 250 U/mL of AlyP1400. The AlyP1400 and tobramycin co-treatment was performed in two manners. The first treatment style, AlyP1400+TOB, AlyP1400 was combined with tobramycin and injected into the biofilms as a whole and incubated for 2 hours. This treatment is referred to as the combinational treatment. In the second one that is referred to as the sequential treatment, AlyP1400→TOB, AlyP1400 was injected into the biofilms and incubated for 2 hours followed by injection of tobramycin and incubation for another 2 hours before collecting cells. Following the treatments, cells were collected at 0, 3, 6, 12 and 24 hours and stained with fixable viability stain 700 (FVS) to detect all cells by emitting red florescence (stained post-fixation), and FVS520 (stained pre-fixation) to allow quantification of membrane-compromised/dead cells within biofilm dispersed communities by emitting green florescence. The live/dead cells were expressed as events per milliliter.

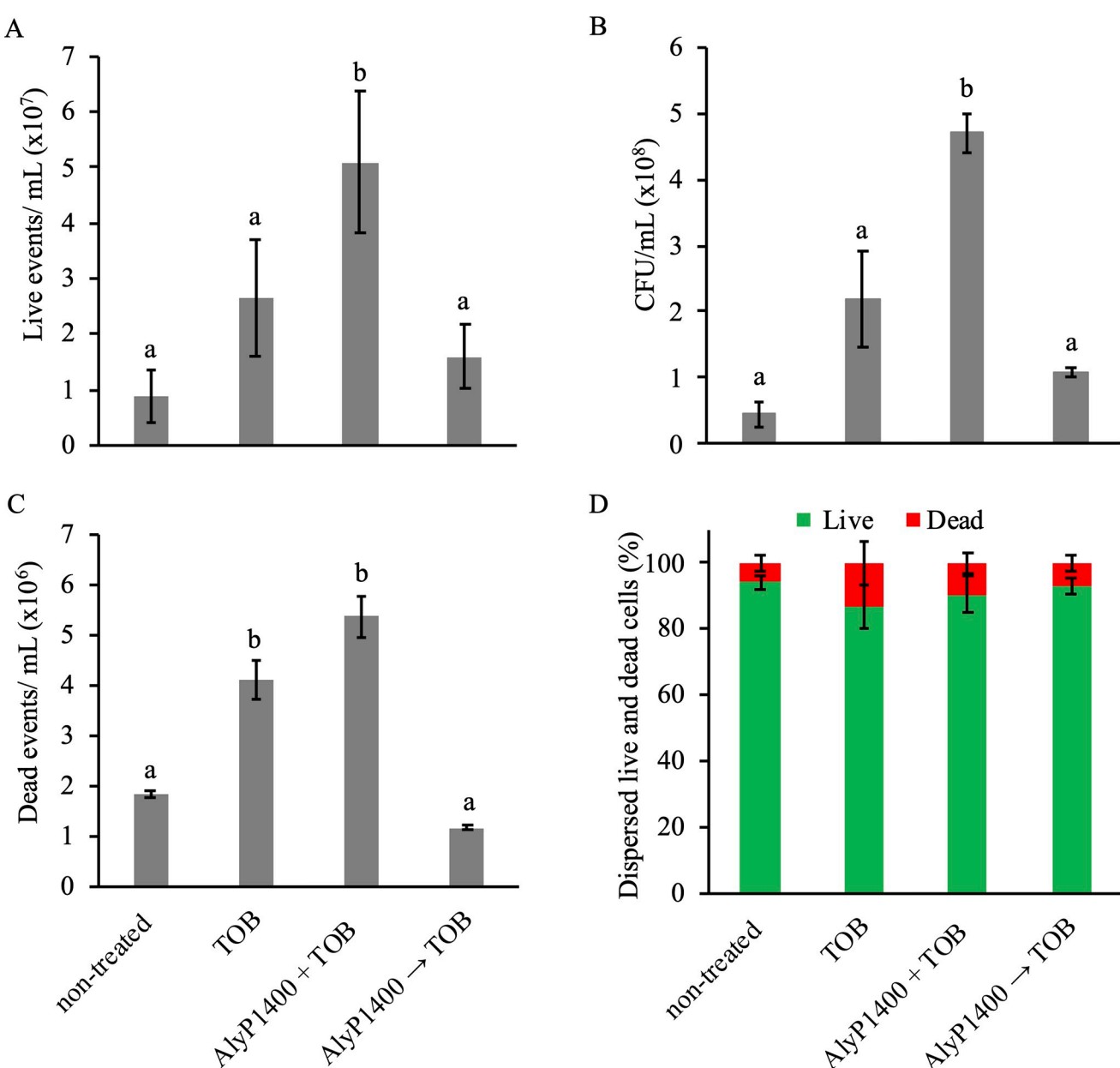

**Fig 1.** *P. aeruginosa* **CF27 biofilm dispersal after treatment with buffer control (non-treated), or 16 μg/mL tobramycin (TOB) or with 250 U/mL AlyP1400 and 16 μg/mL tobramycin in a combinational (AlyP1400+TOB) or sequential (AlyP1400→TOB) manner.** (A) Live events per millilitre (events/mL) as determined using flow cytometric analysis of the cell populations for living cells (events negative for FVS520 and positive for FVS700). (B) The total viable count (CFU/mL) of *P. aeruginosa* CF27 living dispersed biofilm cells. (C) Dead events per millilitre (events/mL) as determined using flow cytometric analysis of the dead populations (events positive for both FVS520 and FVS700). (D) The percentage of dispersed live and dead cells from biofilms. Bars with different letters (a and b) are statistically different (p<0.05, ANOVA two-way with Tukey's multiple comparison test). Data were analyzed for samples collected immediately after the treatments. The standard error of mean of three independent flow cells is indicated by the error bars.

The major observation is that the tobramycin only and AlyP1400+TOB treatments caused the highest rate of dispersal cells at zero hours after the resumption of the flow, whereas the counts of the subsequent time points reached a relatively constant rate for each condition (S2 Fig). Therefore, we focused our comparisons on the time point with the peak of the dispersal cells and the largest differences, which was zero hours after the resumption of the flow (Fig 1).

We saw that the non-treated biofilms had a live dispersal count of $8.72 \times 10^6 \pm 4.75 \times 10^6$ events/mL in the flow cytometric assay (Fig 1A). The live dispersal cell count value was significantly altered after treatment with AlyP1400+TOB, but not with tobramycin alone or AlyP1400→TOB (Fig 1A). The data revealed that the highest dispersed viable cells were triggered by the treatment of AlyP1400+TOB, releasing live cells at $5.1 \times 10^7 \pm 2.72 \times 10^7$ events/ mL (Fig 1A). This trend was also confirmed by the viable count enumerated by CFU/mL. The CFU count from non-treated biofilms revealed the lowest dispersed level of bacterial cells of $5.4 \times 10^7 \pm 1.8 \times 10^7$ CFU/mL, while the counts from biofilms treated with AlyP1400+TOB showed the highest dispersal of $4.7 \times 10^8 \pm 3.0 \times 10^7$ CFU/mL (Fig 1B).

The direct measurement of antibiotic-killed bacterial cells within biofilm is challenging due to the obvious dilemma of counting dead cells. Our established flow cytometric analysis utilizes a fluorescence dye that stains membrane compromised bacterial cells, which can be used to quantify the dead bacterial cell as fluorescence-positive counts. The results showed that both the use of tobramycin alone or AlyP1400+TOB showed statistically significantly increased dispersal of dead cells, but not the AlyP1400→TOB treatment. The highest count of $5.36 \times 10^6 \pm 4 \times 10^6$ counts/mL came from the combinational AlyP1400+TOB treatment (Fig 1C).

We observed that the live cell subpopulation (measured by live cell events divided by the sum of the live cell events plus the dead cell events multiplied by 100) represented the majority of the dispersal cell population, being above 80% under all conditions (Fig 1D). None of the treatments showed significantly different live cell percentage comparing to the non-treated control (Fig 1D).

Unlike the CF27 biofilms, the dispersal of the *P. aeruginosa* non-mucoid strain PAK biofilm was not significantly altered by the addition of AlyP1400 in either combinational or sequential manner to tobramycin (S3 Fig). However, the tobramycin only (TOB) treatment significantly increased the dispersal rate measured by the CFU counting (S3 Fig).

## Alginate lyase-tobramycin combinational treatment leads to distinct expression pattern in antibiotic resistance genes in biofilm dispersed cells

Dispersed cells showed a distinct gene expression profile compared to other stages of biofilm cells and planktonic cells [11]. Our data demonstrated that the combination of AlyP1400 and tobramycin represent a more effective strategy than individual enzyme or antibiotic treatment to disrupt *P. aeruginosa* biofilms indicated by previously quantifying remaining biofilm [16] and currently by the direct measurement of dispersal populations (Fig 1). Because only the combinational but not the sequential treatment showed an enhanced CF27 dispersal, we then set to further characterize how the co-administration of the AlyP1400 (AlyP1400+TOB) could affect the gene expression changes that tobramycin caused on CF27 dispersal cells. Because bacterial transcriptomic profiles are highly unstable and can change rapidly [32,33], we collected dispersal cells from the 96-hour old biofilms grown under dynamic conditions right after the 2-hour treatment with 16 μg/mL tobramycin with or without 250 U/mL of AlyP1400 directly into RNA Protect reagent in order to capture the immediate effects of the synergistic administration of the AlyP1400 and tobramycin on the physiological changes within the dispersed population. To characterize and compare the gene expression profile of dispersed biofilm cells, we initially selected 12 candidate genes to be measured by qRT-PCR in our experimental conditions. These genes included five groups: 1) three genes for alginate production (*algD*, *algR* and *algU*); 2) four efflux genes that are related to antibiotic resistance (*mexB*, *mexF*, *mexY*, and *mexZ*); 3) three genes for secreted enzyme (*lasB*, *ndvB* and *clpX*); 4) the key quorum sensing regulator gene *lasR*; and 5) *bdlA* that codes for a chemosensory protein associated with bacterial biofilm dispersal. Note, that *ndvB* functions in biofilm-specific antibiotic

resistance. We found that, out of the 12 selected genes, *algD*, *clpX*, *mexB* and *mexZ* expression was not abundant enough in our samples, as the qRT-PCR reactions using the primers for these genes did not reach the Ct values. Additionally, the *algU* and *lasR* genes did not show any significant changes between the dispersal samples and planktonic inoculum control.

The data shown in Fig 2 revealed that the dispersal cells exhibited a higher expression level of the *algR*, *bdlA*, *lasB*, *mexF*, *mexY*, and *ndvB* genes when compared to the planktonic control cells that were used as the biofilm cultivation inoculum. While the dispersal cells from the non-treated biofilm and the biofilms that were treated by TOB or AlyP1400+TOB demonstrated similar induction in *lasB* and *algR* genes (Fig 2), there is a significantly higher degree of induction in three of the antibiotic resistance-related genes (*mexY*, *mexF* and *ndvB*) as well as the biofilm dispersion-related gene (*bdlA*) in dispersal cells from biofilm treated with AlyP1400+TOB (Fig 2).

### Alginate lyase-tobramycin combinational treatment increases tobramycin tolerant fraction of biofilm dispersed cells

To test whether the higher induction of antibiotic resistance-related genes in the CF27 dispersal cell population from the combinational treatment (AlyP1400+TOB) (Fig 2) will render the cells more tolerant to subsequent tobramycin exposure, we examined the phenotypic differences in cells released under the treatment conditions. Collected dispersal cells were grown on LB or LB supplemented with sub-MICs of tobramycin. Under our tested growth condition, dispersal cells only survived 1 µg/mL of tobramycin, but not higher concentrations (2–8 µg/ mL). The tobramycin (1 µg/mL) tolerant fraction of the non-treated sample was 6.5%, which was not statistically different than the 5.8% of the planktonic inoculum (Fig 3). Dispersal cells from the TOB-treated biofilms showed a negligibly lower tolerant fraction of 1.5% (Fig 3). However, dispersal cells released from biofilms treated with alginate lyase combined with tobramycin showed a significantly higher fraction of survival (60%) in the presence of 1 µg/mL tobramycin (Fig 3). Meanwhile, although AlyP1400 did not show any effects on PAK biofilm dispersal therefore no gene expression analysis was carried out for this strain, we performed the tobramycin tolerance experiment on PAK dispersal cells. Tolerant fractions under all conditions were under 3% (lower than the non-treated CF27 dispersal cells) (S4 Fig). Not surprisingly, the combinational treatment did not significantly enhance the tolerant fraction compared to the TOB treatment (S4 Fig).

## Discussion

Globally, *P. aeruginosa* is classified as a bacterial species that has high antibiotic resistant activities, therefore it is a dangerous pathogen especially for patients in critical care [34]. Biofilm cells are unreachable by host immune responses and are protected from environmental conditions and antimicrobial agents [35]. Long-term treatment with multiple-antibiotics therapies can be effective for some patients, however this often causes systemic detrimental effect on the patients [36]. There is a pressing need for innovative therapies to treat *P. aeruginosa* biofilm-related infections.

The enzymatic disruption of biofilm by alginate lyases to enhance bactericidal activity of known antibiotics on biofilm cells holds promising therapeutic potentials for treating *P. aeruginosa* infections. While many anti-biofilm studies using alginate lyases focused on the biofilm treatment [12–16], little attention has been paid to the dispersed cells after treatment. The biofilm dispersal cells have a unique intermediate role between the planktonic and biofilm life style, and are highly virulent to the host cells [7,11]. To directly characterize the dispersal cell population and evaluate the effects of the combinational treatment of the alginate lyase

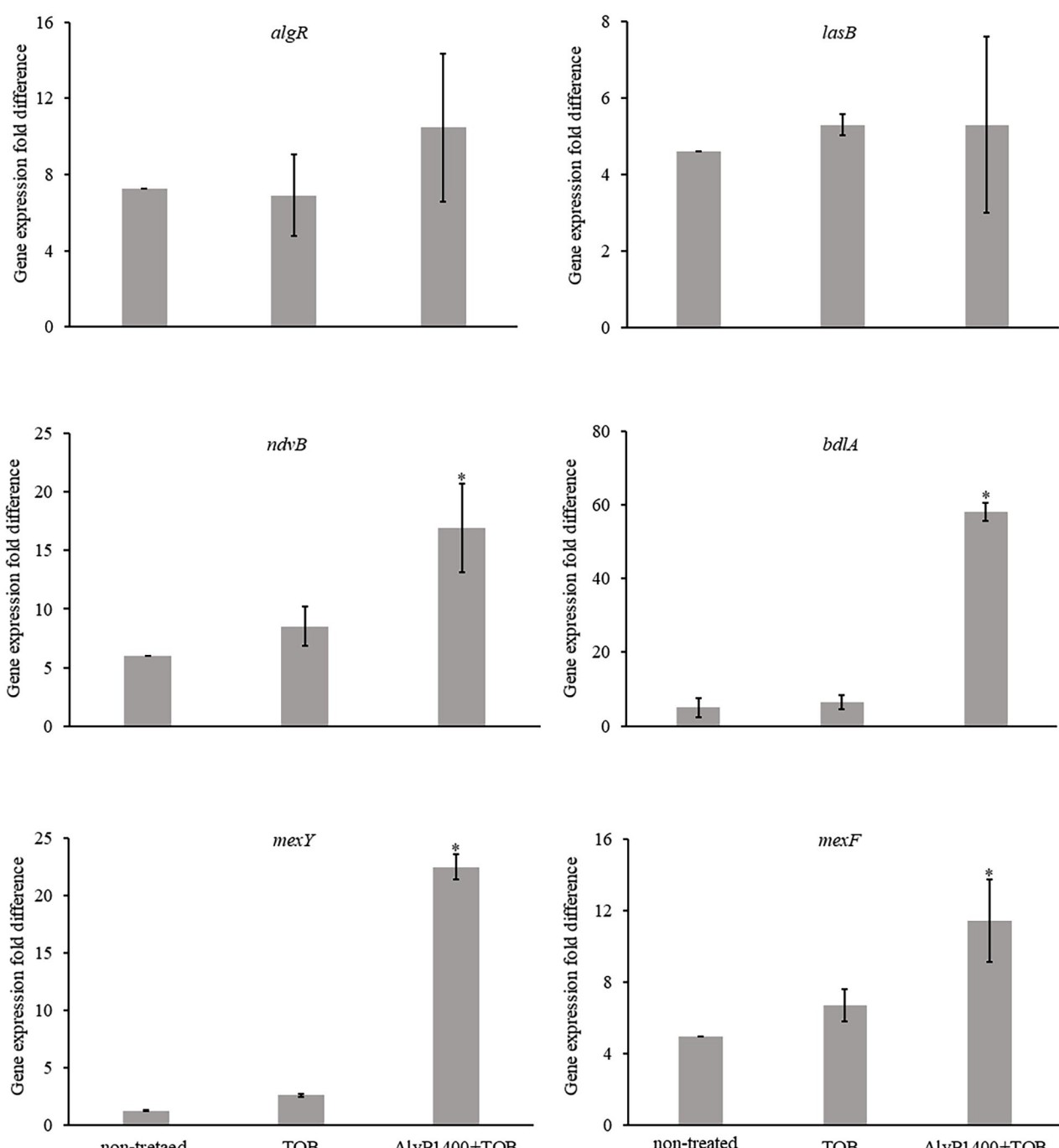

**Fig 2. Relative expression of genes measured by RT-qPCR in CF27 biofilm dispersal cells.** Gene expressions are measured as the gene expression fold difference compared to planktonic control cells, which were used as the inoculum for the biofilm cultivation. Biofilm dispersal cells were collected immediately after a 2-hour treatment with buffer control (non-treated), or 16 μg/mL tobramycin (TOB) or with 250 U/mL AlyP1400 and 16 μg/mL tobramycin (AlyP1400+TOB). The mean and standard deviation are presented for the data from three replicates. Statistical significance was determined using ANOVA one-way analysis with Tukey's test. Tobramycin (TOB). * P < 0.05 compared with both non-treated control and TOB only treatment.

AlyP1400 with tobramycin on the *P. aeruginosa* CF27 dispersed cells, we quantified the biofilm dispersal cells by the flow cytometry technique to assort the viable and dead cells using fluorescent stains to analyze both subpopulations.

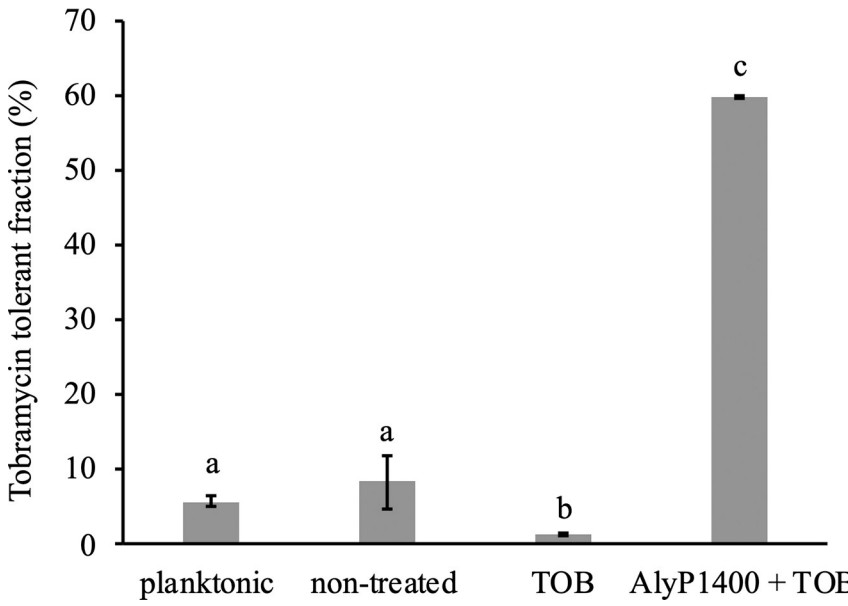

**Fig 3. Tobramycin tolerant fraction (%) of *P. aeruginosa* CF27 planktonic and biofilm dispersal cells to 1 μg/mL of tobramycin.** Bars with different letters (a, b and c) are statistically different (p<0.05, ANOVA one-way with Tukey's multiple comparison test). Data were analyzed for samples collected immediately after a 2-hour treatment with buffer control (non-treated), or 16 μg/mL tobramycin (TOB) or with 250 U/mL AlyP1400 and 16 μg/mL tobramycin (AlyP1400+TOB). The standard error of mean of three independent flow cells is indicated by the error bars.

A vast improvement to CF27 biofilm disruption, leading to the reduction in remaining biofilms (S1 Fig) and increased dispersal cell (both live and dead) count was observed when the biofilm was treated with AlyP1400 in combination with tobramycin (Fig 1). The obvious explanation would be the hydrolytic activity of AlyP1400 caused the dissolution of the biofilm matrix, and thus more cells were released.

Our direct investigation on the CF27 dispersed live and dead cells separated using a flow cytometric analysis revealed that the synergistic effect of alginate lyase and tobramycin in enhancing biofilm cell dispersal was only demonstrated when AlyP1400 was co-administered with TOB in a combinational but not sequential manner (Fig 1A–1C). The fact that there was no enhanced biofilm dispersal after the initial 2-hour treatments with AlyP1400 suggested that the increased dispersal cells caused by the AlyP1400 treatment is likely to happen immediately after biofilm disruption rather than gradually after the disruption and is not long lasting. This is also consistent with the observation on the peak dispersal at time point zero after the resumption of the flow (S2 Fig). Interestingly, live cells represented the majority within the dispersal population even for the tobramycin- and the tobramycin plus AlyP1400-treated samples (Fig 1D). The data suggests that safeguarding approaches need to be taken to prevent potential spreading of infection caused by the use of biofilm disruptive agents to treat bacterial infections in patients. However, this distribution between dead and live cells depends on the concentration of tobramycin used and higher concentrations, like the ones achieved by antibiotic inhalation treatments in the lungs of CF patients, will likely generate a different distribution in live and dead cell percentages.

On the contrary, the combinational treatment of tobramycin plus AlyP1400 did not enhance live cell dispersal from biofilms of PAK, a non-mucoid lab strain of *P. aeruginosa* (S3 Fig). This is not surprising because other exopolysaccharides, but not alginate, are the major components of the biofilm matrix of the non-mucoid *P. aeruginosa* PAK strain [37]. However,

in our previous work, AlyP1400 showed a synergy with tobramycin in reducing the biofilm biomass of another non-mucoid strain PA14. It is worth noticing that alginate was detected as part of the extracellular biofilm matrix of PA14, even though at a much lower level than the one in CF27. Therefore, the biofilm disruptive effect of AlyP1400 can be dependent on the specific exopolysaccharide components of the biofilm matrix, which is variable even within the non-mucoid strains.

A knowledge gap lies in our understanding of how antibiotics-treated biofilms could impact the gene expression profiles of dispersal cells. The dispersed cells and planktonic lifestyle cells were thought to be similar and both were assumed to be eradicated easier than the biofilm cells [7]. However, a recent study induced cell dispersal from *P. aeruginosa* biofilms by reducing cellular c-di-GMP levels via chemical or genetical approaches and showed distinct gene expression profiles for induced dispersal cells, which were associated with their higher virulence against immune cells than their planktonic counterparts [11]. The authors proposed that the process of dispersion conferred "protection-in-advance mechanisms," which enhanced virulence in biofilm dispersal cells to cope with environmental insults [11]. To our knowledge, this study is the first to examine the effects of tobramycin alone and tobramycin combined with AlyP1400 on the gene expression patterns of the biofilm dispersal cells.

Despite the fact that the mucoid strain CF27 is a clinical isolate, whose genomic information is not complete to guide a sophisticated gene expression analysis, we were able to carry out the expression analysis of 12 genes and successfully detect the expression differences in several virulence- or antibiotic resistance-related genes, including *algR*, *bdlA*, *lasB*, *mexF*, *mexY*, and *ndvB*. AlgR is a regulatory protein that functions in many different pathways, most notably the alginate biosynthesis pathway, but also include production pathways of cyanide, rhamnolipids, and LPS [38–40]. BdlA is a chemotaxis regulator essential for biofilm dispersion in *P. aeruginosa*. The gene *lasB* encodes for elastase B, a major type II-secreted virulence factor [41], whose expression and secretion are under the control of the quorum system. Resistance-nodulation-cell division (RND) multidrug efflux transporter genes (*mexF* and *mexY*) belong to the efflux pump group that are some of the main contributors to the antibiotic resistance in *P. aeruginosa*. The gene *ndvB* encodes for a glucosyltransferase enzyme, promoting the production of periplasmic β-(1→3) cyclic glucans that can enhance aminoglycoside resistance through disposal of the antibiotics from their cellular target.

Our results showed that the CF27 dispersal cells with or without any treatment induced all six genes compared to the planktonic control cells (Fig 2). The induction of the multidrug efflux RND (*mexF* and *mexY*) genes in the dispersal cell population seen in this study supports the observation from a previous publication that showed dispersal cells exhibited a higher expression level of the RND genes, such as *mexG*, *mexH* and *mexI* [11]. The virulence-related genes (*algR* and *lasB*), were induced to similar levels in dispersal cells regardless of the treatment causing the dispersion (Fig 2), suggesting that these genes can be part of the characteristics of dispersal cells in general. It is believed that *P. aeruginosa* expresses virulence factors, including pyocyanin, elastase, and rhamnolipids for maximal invasiveness of *P. aeruginosa* in hosts [42], which can be required for them to re-establish biofilm colonies within the host environment. The increases in the *lasB* and *algR* genes in the dispersal cells compared to the planktonic cells suggest that these genes may contribute to the preparation of dispersal cells for re-establishment of succeeding biofilm. It is worth noting that a previous study showed that *lasB* was suppressed in dispersal cells compared to the planktonic cells [43]. One possible explanation for this discrepancy between our data is that the planktonic cell control in our study was the minimal medium washed *P. aeruginosa* cells used for biofilm cultivation inoculum, whereas the one used in Li et al. [43] was the overnight culture of *P. aeruginosa*, which should have the maximum induction of *lasB* [44]. The drastic dissimilar basal levels of *lasB* in

our different controls could potentially reconcile the differences in its relative changes in the dispersal cells.

Our data showed that tobramycin alone did not significantly change the induction in the selected virulence- and antibiotic resistance-related genes in the CF27 dispersal cells (Fig 2). Interestingly, the combinational treatment of tobramycin and AlyP1400 further induced three of the antibiotic resistance-related genes (*mexF*, *mexY*, and *ndvB*), as well as the biofilm dispersal gene *bdlA*, but not the virulence-related genes (*algR* and *lasB*), to a higher degree (Fig 2). A previous report showed the biofilms of *P. aeruginosa ndvB* mutants exhibited increased sensitivity to the aminoglycoside tobramycin [45]. These findings together revealed an important role of *ndvB* in mediating biofilm and dispersal cell antibiotic resistance. A similar expression pattern was detected for the *bdlA* gene for cells released from biofilms treated with AlyP1400 combined with tobramycin (Fig 2), which was consistent with the increased dispersal cell count that was observed when the biofilm was treated with the combinational treatment (Fig 1). BdlA is essential for the *P. aeruginosa* biofilm dispersal and its expression was shown to be induced in dispersal cells [46]. Our finding that BdlA is further induced by the combinational treatment of alginate lyase and tobramycin suggests there is likely a regulatory mechanism triggered by this combined effect to amplify the dispersal signal.

Our tobramycin tolerant assays showed a clear survival advantage of the CF27 dispersal cells released by the AlyP1400 and tobramycin combinational treatment (Fig 3), in correlation to their significantly higher expression of antibiotic resistance-related genes, especially *mexY* (Fig 2). The MexXY-OprM operon system comprises a cytoplasmic membrane antibiotic-proton antiporter (MexY), an outer membrane porin (OprM) and a periplasmic membrane fusion protein, that facilitates passage of the substrate across the outer membrane, which joins the membrane-associated components together (MexX) [47]. MexXY-OprM pump is often associated with active efflux of aminoglycosides and overexpressed in CF isolates of *P. aeruginosa* [48]. Due to the correlation between *mexX* and *mexY* mRNA expression and MexXY protein is predominantly transcriptionally regulated [49], we opted to analyse *mexY* expression as marker for MexXY protein expression. The overexpression of *mexY* was dramatic in dispersal cells released from the combinational treatment compared to the dispersal cells from non-treated biofilms. This genotypic characteristic was clearly positively associated with the higher fraction of dispersal cells surviving subsequent exposure to tobramycin. Overall, our data of gene induction patterns and associated phenotypic survival to tobramycin support a model that the tobramycin has enhanced diffusion activity through the alginate layers after hydrolysis by AlyP1400 took place within mucoid *P. aeruginosa* biofilms and the dispersal cells released from biofilms by the combination of tobramycin with AlyP1400 have a higher magnitude of tolerance towards tobramycin-triggered antibiotic stress response. On the contrary, the tobramycin tolerant fractions of non-treated and treated dispersal cells as well as planktonic cells of PAK were all below 3%, suggesting a different resistance-associated genetic background of this strain.

In conclusion, our work sheds new light on the viability and gene expression statuses of the bacterial pathogen cells released from the biofilm (Fig 4). Our findings revealed that large quantities and percentages of the dispersal population were viable, and were expressing genes related with conferring enhanced antibiotic-resistance and biofilm dispersal. These findings raise concerns about biofilm disruption approaches, and in turn, this knowledge can better guide the future design of therapeutic strategies utilizing the combination of biofilm disruptive agents, such as our AlyP1400, with antibiotics to treat *P. aeruginosa* lung infection in CF patients.

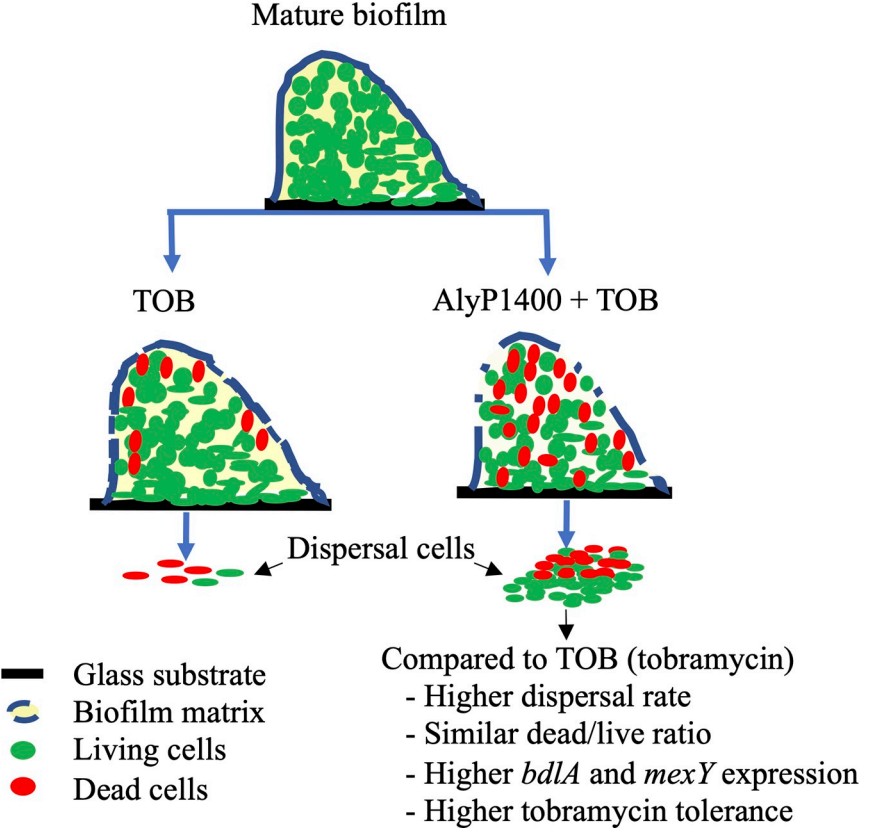

**Fig 4. Synergistic effect of tobramycin with AlyP1400 on *P. aeruginosa* CF27 biofilm dispersal cells.**

## Supporting information

**S1 Fig. Images of *P. aeruginosa* CF27 biofilms grown in flow cell systems after treatment with 16 μg/mL tobramycin (TOB) individually or with 250 U/mL AlyP1400 compared to non-treated, control.** The residual biofilms were visualized with Confocal Laser Scanning Microscopy (CLSM) after staining with Syto 61 Red, (stained all cells red) and Sytox Green, (stained compromised membrane and /or dead cells green) using magnification power 63X. The yellow fluorescence represented merging of both red and green channels in the merged panel. Images are representatives from three independent replicates with 20 μm bars. (TIF)

**S2 Fig. *P. aeruginosa* CF27 biofilm dispersal at 0, 3, 6, 12, and 24 hours after treatment with, buffer control (non-treated), or 16 μg/mL tobramycin (TOB) individually or with 250 U/mL AlyP1400 (AlyP1400+TOB).** Sequential treatment (AlyP1400→TOB) data were only acquired for 0 h time point, therefore were not included in this figure. (A) Live events per millilitre (events/mL) as determined using flow cytometric analysis of the cell populations for living cells (events negative for FVS520 and positive for FVS700). (B) The total viable count (CFU/mL) of *P. aeruginosa* CF27 living dispersed biofilm cells. (C) Dead events per millilitre (events/mL) as determined using flow cytometric analysis of the dead populations (events positive for both FVS520 and FVS700). Statistical significance was determined using ANOVA two-way analysis with Tukey's test multiple comparison test. * P < 0.05, ** P < 0.01 compared with non-treated control. The standard error of mean of three independent flow cells is

indicated by the error bars.
(TIF)

**S3 Fig. The total viable count (CFU/mL) of *P. aeruginosa* PAK living dispersed biofilm cells after treatment with 16 μg/mL tobramycin (TOB) or with 250 U/mL AlyP1400 and 16 μg/mL tobramycin in a combinational (AlyP1400+TOB) or sequential (AlyP1400→TOB) manner.** Bars with different letters (a and b) are statistically different (p<0.05, ANOVA two-way with Tukey's multiple comparison test). Data were analyzed for samples collected immediately after the treatments. The standard error of mean of three independent flow cells is indicated by the error bars.
(TIF)

**S4 Fig. Tobramycin tolerant fraction (%) of *P. aeruginosa* PAK planktonic and biofilm dispersal cells to 0.5 μg/mL of tobramycin.** Bars with different letters (a and b) are statistically different (p<0.05, ANOVA one-way with Tukey's multiple comparison test). Data were analyzed for samples collected immediately after a 2-h treatment with buffer control (non-treated), or 16 μg/mL tobramycin (TOB) or with 250 U/mL AlyP1400 and 16 μg/mL tobramycin (AlyP1400+TOB). The standard error of mean of three independent flow cells is indicated by the error bars.
(TIF)

## Author Contributions

**Conceptualization:** Said M. Daboor, Zhenyu Cheng.

**Data curation:** Renee Raudonis.

**Formal analysis:** Said M. Daboor, Renee Raudonis.

**Funding acquisition:** Zhenyu Cheng.

**Methodology:** Renee Raudonis.

**Supervision:** Zhenyu Cheng.

**Writing – original draft:** Said M. Daboor, Renee Raudonis, Zhenyu Cheng.

**Writing – review & editing:** Said M. Daboor, Renee Raudonis, Zhenyu Cheng.

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
