## [Decision Letter · Decision Letter 0]

4 May 2021

PONE-D-21-07365

Characterizations of the viability and gene expression of dispersal cells from Pseudomonas aeruginosa biofilms released by alginate lyase and tobramycin

PLOS ONE

Dear Dr. Cheng,

Thank you for submitting your manuscript to PLOS ONE. After careful consideration, we feel that it has merit but does not fully meet PLOS ONE’s publication criteria as it currently stands. Therefore, we invite you to submit a revised version of the manuscript that addresses the points raised during the review process.

We look forward to receiving your revised manuscript.

Kind regards,

Abdelwahab Omri, Pharm B, Ph.D

Academic Editor

PLOS ONE

Journal Requirements:

2. In your Methods section, please provide additional details regarding the cell lines used in your study and ensure you have described the source. For more information regarding PLOS' policy on materials sharing and reporting, see https://journals.plos.org/plosone/s/materials-and-software-sharing#loc-sharing-materials, and for more information on PLOS ONE's guidelines for research using cell lines, see https://journals.plos.org/plosone/s/submission-guidelines#loc-cell-lines.

[This project is supported by the Cystic Fibrosis Canada Marsha Morton Early Career Investigator award. This project was also supported by a Nova Scotia Health Research Foundation Establishment Grantand a Canadian Institutes of Health Research Project Grant (PJT165970)to Z.C.]

 [The funders had no role in study design, data collection and analysis, decision to publish, or preparation of the manuscript.]

Reviewers' comments:

Reviewer's Responses to Questions

**Comments to the Author**

1. Is the manuscript technically sound, and do the data support the conclusions?

Reviewer #1: Yes

Reviewer #2: No

2. Has the statistical analysis been performed appropriately and rigorously? 

Reviewer #1: Yes

Reviewer #2: No

3. Have the authors made all data underlying the findings in their manuscript fully available?

Reviewer #1: Yes

Reviewer #2: Yes

4. Is the manuscript presented in an intelligible fashion and written in standard English?

Reviewer #1: Yes

Reviewer #2: Yes

5. Review Comments to the Author

Reviewer #1: Main findings of the study:

The work reports the characterization, namely in terms of cell viability and gene expression profiles, of cells dispersed by P. aeruginosa biofilms upon treatment with tobramycin (TOB) alone and in combination with alginate lyase (AlyP1400). First, the authors used flow cytometry to estimate the biofilm released cells and to discriminate them into live and dead. Second, transcriptomics was performed in released biofilm cells, where the expression of 12 genes related to virulence or resistance was analyzed. Afterwards (although not mentioned in the abstract), the survival ability of biofilm released cells to a subsequent TOB treatment was addressed by plating the biofilm dispersal cells onto LB agar medium supplemented with TOB at sub-MICs. This is a continuing study of a previous work where the authors demonstrated a reduction in the biomass and degradation of alginate within the EPS matrix of P. aeruginosa biofilms, thus enhancing the antibiofilm activity of TOB.

The findings in this present study showed that TOB+AlyP1400 could increase the number of biofilm-released cells, which were apparently stained green (live) in their majority. At the gene expression level, these cells (when treated with TOB+AlyP1400) had upregulated 4 out of 12 virulence- or resistance-associated genes (bdlA, mexF, mexY, ndvB). Survival studies displayed a significant high TOB tolerant fraction of cells (60%) upon treatment with TOB+AlyP1400.

Limitations and strengths:

Although the work does not fall into an original topic, the paper shows relevant results regarding the use of combinatorial treatment employing biofilm disruptive agents and conventional antibiotics. There are, in fact, multiple studies reporting synergistic activity of alginate lyase with many antibiotic agents, including TOB. However, this study focuses on the characterization of biofilm-released cells upon treatment with alginate lyase+TOB. These cells should not be overlooked and the results sounds the alarm for potential likelihood of severe reinfections by individual cells treated with this therapeutic approach. Overall work and its goal are logically presented. The introduction defines clearly the rationale of the work, considering the fact that the authors have already reported the potentiality of alginate lyase against P. aeruginosa biofilm biomass and its ability to boost TOB action towards these infections.

The methods used in the work are generally well detailed and can be reproduced by others. The results and data interpretation are clearly presented.

Nevertheless, in some parts, it becomes difficult to read and follow. The paper contains some errors in the scientific terminology and should be carefully reviewed and rewritten in some parts.

There are several issues that need to be addressed before publication:

1. Alginate formation by Pseudomonas aeruginosa is an important bacterial resistance mechanism, by completely preventing or blocking the diffusion of various antibiotics, namely aminoglycosides. While applying alginate lyase, this will disrupt biofilm matrix and allow TOB to penetrate and target biofilm-encased cells. I am wondering if the dispersal of biofilm cells (which are mostly live in these study) happens immediately after biofilm disruption by alginate lyase or the cells will be slowly being dispersed to the environmental milieu… It would be interesting to evaluate also co-administration of AlyP1400 followed by TOB and compare the results with that obtained in this study. Moreover, what happens to the cells residing in the biofilm? Are they live or dead?

2. In this work, the authors used a mucoid CF isolated strain (CF27) to grow biofilms for 96 h, which were then treated (with TOB alone or TOB+AlyP1400) and further tested on their viability, genetic expression and survival upon a second TOB single treatment. However, I am concerned whether this is representative of a major population of isolated strains (i.e. if other CF isolate were used, their behavior would be similar to this one of the CF27? Moreover, why do not include a reference strain, maybe P. aeruginosa ATCC 27853, because it is well known that isolates may have resistance-associated backgrounds that may interfere with later results?

3. Concerning the age of the biofilm, why the authors have selected 96h to have old mature biofilms? What is the advantage of using biofilms at 96 h instead of 24 or even 48h?

4. Why forming biofilms on flow dynamic cell systems having glass as substratum? What is the advantage of biofilms formed on these devices and biofilms formed on reproducible microtiter plates, for example?

5. If not previously addressed, it would be interesting to evaluate the effect of the combination of the alginate lyase and TOB through the FIC outcome (checkerboard assay) against P. aeruginosa.

6. I think having a schematic representation of the concept of the study would be valuable.

Reviewer #2: The manuscript by Daboor et al., aims to examine the viability and phenotype of chemically-induced dispersal of cells from Pseudomonas aeruginosa biofilms. Using a previously characterized clinical isolate, the authors confirm previous dispersal results and attempt to quantify the viable cellular material using fluorescent markers for viability . Next they studied the expression of a limited number of genes related to antibiotic resistance and virulence. Their findings indicate that some of these genes are increased in expression when cells are exposed to antimicrobials and dispersal agents.

The overall goal of the manuscript, which is to study the mechanics of anti-alg based treatments in conjunction with tobramycin are appealing. There is a lack of work in the determining the mechanisms and more importantly the phenotypes related to the process of biofilm dispersal. However, the authors results are do not convincingly make the case that they are actually, or accurately detecting this process. Much technical insight/information and data are required to support their observations, specifically with respect to the how dispersal is quantified. Below, several major areas that require addressing are discussed. The gene expression data is intriguing and are the highlight of the paper, but on their own seem superficial and not well supported.

Comments:

- The description of how displeased cells were measured relative to the initial biofilm biomass is poorly described. How consistent were each biofilm in prior to treatment in terms of number of cells (CFU/ml) and does the variability affect the dispersal over time?

- Starting at line 212, the authors state that maximal dispersal occurs at zero hours - how was this determined and where is it described in Figure 1? Can the data be presented so that the time dependence is clearly discernable? The A, B, C designation to indicate the level of error in Figure 1 is also confusing as the panels are labelled similarly.

In Figure 1D, the differences observed with the with the combination treatment of AlyP1400+Tob are not convincingly different than non-treated or Tob treated cells.

6. PLOS authors have the option to publish the peer review history of their article (what does this mean?). If published, this will include your full peer review and any attached files.

Reviewer #1: No

Reviewer #2: No

---

## [Editor Report · Decision Letter 1]

11 Oct 2021

Characterizations of the viability and gene expression of dispersal cells from Pseudomonas aeruginosa biofilms released by alginate lyase and tobramycin

PONE-D-21-07365R1

Dear Dr. ,Zhenyu Cheng,

We’re pleased to inform you that your manuscript has been judged scientifically suitable for publication and will be formally accepted for publication once it meets all outstanding technical requirements.

Kind regards,

Abdelwahab Omri, Pharm B, Ph.D

Academic Editor

PLOS ONE

---

## [Editor Report · Acceptance letter]

14 Oct 2021

PONE-D-21-07365R1 

Characterizations of the viability and gene expression of dispersal cells from *Pseudomonas aeruginosa* biofilms released by alginate lyase and tobramycin 

Dear Dr. Cheng:

I'm pleased to inform you that your manuscript has been deemed suitable for publication in PLOS ONE. Congratulations! Your manuscript is now with our production department. 

Kind regards, 

on behalf of

Dr. Abdelwahab Omri 

Academic Editor

PLOS ONE